# Sex differences in pulmonary function parameters among Indigenous Australians with and without chronic airway disease

**Subash S. Heraganahally**[1,2,3]*, **Timothy Howarth**[3,4], **Lisa Sorger**[5], **Helmi Ben Saad**[6,7]

**1** Department of Respiratory and Sleep Medicine, Royal Darwin Hospital, Darwin, Northern Territory, Australia, **2** Flinders University - College of Medicine and Public Health, Adelaide, South Australia, Australia, **3** Darwin Respiratory and Sleep Health, Darwin Private Hospital, Darwin, Northern Territory, Australia, **4** College of Health and Human Sciences, Charles Darwin University, Darwin, Northern Territory, Australia, **5** Department of Medical Imaging, Royal Darwin Hospital, Darwin, Northern Territory, Australia, **6** Faculty of Medicine of Sousse, Laboratory of Physiology, University of Sousse, Sousse, Tunisia, **7** Farhat HACHED Hospital, Research Laboratory "Heart Failure, LR12SP09", University of Sousse, Sousse, Tunisia

* hssubhashcmc@hotmail.com, Subash.heraganahally@nt.gov.au

**Data Availability Statement:** Data used for this study is accessible from the following DOI: 10.25913/VF9Z-3J59.

## Abstract

### Background

Studies assessing normative values and sex differences in pulmonary function test parameters (PFTPs) among Indigenous populations are sparse.

### Methods

PFTPs were compared between male and female Indigenous Australian adults with and without chest radiologically proven chronic airway diseases (CADs).

### Results

485 adults (56% were female) with no significant difference in age, body mass index or smoking status between sexes were included. Females displayed a higher prevalence of radiology without CADs compared to males (66 vs. 52%, respectively). Among patients without CADs, after adjustment for age, stature and smoking, males displayed significantly higher absolute values of Forced Vital Capacity (FVC) (mean difference, 0.41L (0.21,0.62), p<0.001) and Forced Expiratory Volume in one second ($FEV_1$) (mean difference 0.27L (0.07,0.47), p<0.001), with no significant difference in $FEV_1$/FVC ratio (mean difference -0.02 (-0.06, 0.02), p = 0.174). Male and female patients with radiologically proven CADs demonstrated lower $FEV_1$/FVC values. However, compared to females, males showed significantly greater reductions in pre- [-0.53 (-0.74, -0.32) vs. -0.29 (-0.42, -0.16), p = 0.045] and post- [-0.51 (-0.72, -0.3) vs. -0.27 (-0.39, -0.14), p = 0.049] bronchodilator $FEV_1$.

**Funding:** The author(s) received no specific funding for this work.

**Competing interests:** The authors have declared that no competing interests exist.

## Conclusions

There are significant sex differences in the PFTPs among Indigenous Australians. Recognising these differences may be of value in the accurate diagnosis, management, monitoring and prognostication of CADs in this population.

## Introduction

Pulmonary function tests (PFTs) are crucial in the diagnosis, management, monitoring and prognostications of several respiratory disorders [1]. Individuals' PFT parameters (PFTPs) depend not only on their age, height and weight, but also sex [2–4]. PFTPs provide evidence regarding the nature and severity of respiratory conditions. Moreover, the PFT patterns displayed in the presence of respiratory disease may differ between males and females [5–7]. Earlier published reports have demonstrated sex differences in PFTPs among non-Indigenous ethnic populations [8–10].

Chronic respiratory conditions among adult Indigenous populations are highly prevalent worldwide, especially among Indigenous people living in the English-speaking Organisation for Economic Co-operation and Development countries, including Australia [11]. In the Australian context, approximately 3.3% of the population self-identifies as Indigenous Australians [12]. Indigenous Australians are noted to have a higher prevalence of respiratory disorders, in particular, chronic obstructive pulmonary disease (COPD) and bronchiectasis compared to non-Indigenous Australians, and even more so among those living in the Northern Territory (NT) of Australia [13–16]. Despite literature evidence suggesting that chronic airway diseases (CADs) are highly prevalent in the Indigenous population with significant effects on morbidity and mortality, there appears a substantial gap in knowledge regarding the normative reference PFT values for Indigenous Australians [17]. The limited data published examining PFTPs among adult Indigenous Australians suggest that Forced Vital Capacity (FVC) and Forced Expiratory Volume in one second ($FEV_1$) values are lower in comparison to non-Indigenous Caucasian counterparts, while the $FEV_1$/FVC ratio is nearly preserved [18, 19]. However, evidence in the literature examining if there are any sex differences in PFTPs among the Indigenous Australian population has not been reported in the past, nor if there is a differential effect of CADs on PFTPs between the sexes.

Hence, it may be meaningful to understand sex differences in PFTPs in order to accurately diagnose and manage chronic respiratory conditions. Therefore, the aim of this study was to evaluate and to compare PFTPs according to sex amongst adult Indigenous Australian patients with and without radiological evidence of CADs in the Top End Health Service (TEHS) region of the NT of Australia.

## Methods

### Study design and setting

This retrospective study was conducted at the respiratory and sleep service based at the Royal Darwin Hospital and at Darwin respiratory and sleep health, based at the Darwin Private Hospital in the TEHS region, NT of Australia. Study participants included were Indigenous Australian adults living in the TEHS region, who underwent a PFT between 2012 and 2020. For health care delivery, the TEHS is the major service provider for the NT of Australia servicing about 249,220 people, of whom 30% are of First Nations Indigenous Australian descent [11]. Patients were referred for a PFT by a primary health practitioner, respiratory specialist or

other specialist physician as a part of routine clinical care. This study is a part of a larger project assessing factors influencing and implications of PFTPs in Indigenous Australians and was approved by the Human Research Ethics Committee. The ethics committee waived the requirement for informed consent for this study.

## Study participants—Inclusion and exclusion criteria

Indigenous Australian patients who underwent PFTs between 2012–2020 and had a chest radiography available (Chest X-Ray and computed tomography (CT)) to assess the presence/absence of any radiological pulmonary abnormalities (specifically demonstrating radiographic evidence of COPD/emphysema, bronchiectasis or a combination of both COPD & bronchiectasis) were included in the analysis. Patients' PFTs that were assessed as not acceptable for session quality were excluded.

## Clinical data collection

As per standard protocol, patients age, sex, height and weight were recorded. Body mass index (BMI) was calculated. Smoking history was recorded to identify current-, past- or never-smokers, and to quantify the pack-years of smoking. Further details in relation to methods and settings are available from previous reports from our centre [18, 19].

## Pulmonary function tests

All PFTs were performed according to the American thoracic and the European respiratory societies guidelines/recommendations, including calibration of equipment and quality assurance [20] and as detailed in our previous reports [18, 19]. The PFTs were performed via a portable single-breath diffusing capacity device (EasyOne Pro®, ndd Medical Technologies) device [21]. Only PFTs graded as acceptable for quality, as assessed individually by volume-time and flow-volume graphs for session quality, were included in this study.

As per usual protocol, all patients undergoing elective PFTs were instructed to refrain from smoking for at least two to four hours prior to spirometry testing and to avoid using airway directed inhaled bronchodilator therapy during the preceding 4–12 hours. As per usual practice in our centre and according to the Global Initiative for Chronic Obstructive Lung Disease criteria used in earlier Burden of Obstructive Lung Disease studies, bronchodilator responsiveness (BDR) for spirometry parameters were assessed 15–20 minutes after inhalation of 400 μg of salbutamol via a spacer [22].

In our centre and in the absence of specific PFT reference values for the Indigenous Australian population, the predicted normative values for PFTs were calculated using the National Health and Nutrition Examination Survey Caucasian reference set's (NHANES-III, no ethnic correction was used) [23]. The following pre- and post- bronchodilator PFTPs were determined: $FEV_1$ (L, %), FVC (L, %), and $FEV_1$/FVC (absolute value, %).

## Statistical analysis

Continuous data were checked for normality graphically via histograms. Height, weight, BMI, and smoking pack-years were identified to be non-parametrically distributed. Non-parametric data were presented as medians (interquartile ranges (IQRs)), normal data as means (95% confidence intervals (CI)) and categorical data as numbers (%). Demographic data were compared between sexes via two-tailed proportions z-test for categorical data, two-tailed Students t-test for normally distributed data, and equality of medians test for non-parametric data. PFTPs were split by those with and without radiological finding of CADs, and tested between sexes

via two-tailed Students t-tests. Multivariate linear regression models were developed with and without CADs to adjust for the effects of age, height, weight and smoking status on PFTPs differences between sexes, with results reported as β-coefficients (95% CIs). Multivariate regression models were also developed for female and male patients, adjusting for age, height, weight and smoking status to define the effects of radiology abnormality for each sex, reporting results as β-coefficients (95% CI). Post-estimation equations were used to determine the equality of the β-coefficient for 'CADs' between sexes. All analyses were conducted in STATA IC 15, and α set to p = 0.05 throughout.

## Sample size

As the study was a retrospective study no prior sample size estimation was conducted, however, post-hoc power analysis was conducted for testing of mean differences between females and males for pre- bronchodilator values of FVC, $FEV_1$ and $FEV_1$/FVC (Table 1).

# Results

## Clinical and demographics data

Of the 1350 patients examined during the study period, 485 (271 (56%) female) were eligible to be included in this study. There was no significant difference in age, overall BMI or the proportion of patients who were current- or former- smokers between females and males, though a greater proportion of females reported never smoking (17 vs. 10%, p = 0.046) (Table 2). A significantly greater proportion of females had radiographic findings without CADs compared to males (66 vs. 52%, p = 0.002), and lower proportion of females displayed combined COPD & Bronchiectasis (4 vs. 13%, p = 0.001).

## Pulmonary function test results

Among patients without radiological evidence of CADs, females displayed significantly reduced absolute values for FVC and $FEV_1$ both pre- and post- bronchodilator. However, the $FEV_1$/FVC ratio did not significantly differ between sexes, and the percentage predicted values for each parameter were comparable (Table 3). Among patients with radiological evidence of CADs, females again displayed significantly reduced absolute values for FVC and $FEV_1$ both pre- and post- bronchodilator; and the percentage predicted values for each of these parameters were comparable. The $FEV_1$/FVC ratio was significantly reduced in males compared to females for both absolute, and percent predicted values, pre- and post- bronchodilator with CADs. No significant difference in BDR was noted between sexes with or without radiographic abnormalities.

## Multivariate linear regression analysis

Following adjustment for demographic factors (age, height, weight, and smoking status) in multivariate linear regression, significant sex differences remain for patients without radiographical evidence of CADs for absolute values pre- and post- bronchodilator for FVC (mean

**Table 1. Post hoc power analysis of mean absolute values (pre- bronchodilator) for differences between sexes.**

| Lung function parameters | Females (mean ± SD) (n = 271) | Males (mean ± SD) (n = 214) | Power |
|---|---|---|---|
| FVC (L) | 2.008 (0.631) | 2.771 (0.836) | 1.000 |
| $FEV_1$ (L) | 1.492 (0.611) | 1.942 (0.851) | 1.000 |
| $FEV_1$/FVC | 0.728 (0.131) | 0.678 (0.154) | 0.971 |

**Table 2. Demographic and clinical data for study patients.**

| Clinical parameters | Variables | Total sample (n = 485) | Female (n = 271) | Male (n = 214) | p-value |
|---|---|---|---|---|---|
| Body stature and corpulence | Age (years) | 50.95 (49.85, 52.05) | 51.19 (49.67, 52.71) | 50.65 (49.05, 52.24) | 0.631 |
| | Height^ (m) | 1.65 (1.6, 1.72) | 1.6 (1.57, 1.65) | 1.72 (1.68, 1.76) | <0.001* |
| | Weight^ (kg) | 74 (59, 90) | 71 (58, 87) | 76 (61, 94) | 0.027* |
| | BMI^ (kg/m$^2$) | 26.82 (21.89, 32.08) | 27.4 (22.99, 32.85) | 25.54 (20.62, 31.53) | 0.119 |
| | Underweight (BMI < 18.5 kg/m$^2$) | 57 (12) | 24 (9) | 33 (15) | 0.029* |
| | Normal weight (BMI: 18.5–24.9 kg/m$^2$) | 143 (30) | 74 (28) | 69 (32) | 0.269 |
| | Overweight (BMI: 25.0–29.9 kg/m$^2$) | 113 (23) | 67 (25) | 46 (21) | 0.367 |
| | Obesity (BMI ≥ 30.0 kg/m$^2$) | 169 (35) | 103 (38) | 66 (31) | 0.083 |
| Smoking status | Current smoker | 245 (51) | 134 (50) | 111 (53) | 0.571 |
| | Former smoker | 167 (35) | 89 (33) | 78 (37) | 0.392 |
| | Never smoker | 67 (14) | 45 (17) | 22 (10) | 0.046* |
| | Pack years^ | 18 (4.05, 37.5) | 12 (3.75, 32) | 20 (5, 37.5) | 0.265 |
| Radiography | Chest CT Scan available | 212 (72) | 115 (69) | 97 (77) | 0.124 |
| | No CADs | 290 (60) | 179 (66) | 111 (52) | 0.002* |
| | COPD | 111 (23) | 54 (20) | 57 (27) | 0.081 |
| | Bronchiectasis | 45 (9) | 26 (10) | 19 (9) | 0.787 |
| | Combined COPD & Bronchiectasis | 39 (8) | 12 (4) | 27 (13) | 0.001* |
| | Total CAD findings | 195 (40) | 92 (34) | 103 (48) | 0.002* |

Continuous and categorical data were reported as mean (95% CI) and number (%), respectively.

^Non-parametric data were reported as median (IQR).

p-values derived from 2-tailed Students t-test (normally distributed data), equality of medians test (non-parametric data) or 2-tailed proportions z-test (categorical data).

*Denotes statistically significant association (p<0.05).

**Abbreviations**: **BMI**, Body mass index; **CAD**, Chronic airways disease; **COPD**, Chronic obstructive pulmonary disease; **CT**, Computed tomography; **CXR**, Chest x-ray; **PFT**, Pulmonary function test.

difference without CAD's 0.41 (0.21, 0.62) & 0.4 (0.2, 0.59)) and FEV$_1$ (0.27 (0.07, 0.47) & 0.28 (0.08, 0.48)) (Table 4). Among patients with radiological abnormalities for CADs, the sex differences remained statistically significant for both pre- and post- bronchodilator for FVC (0.33 (0.11, 0.56) & 0.36 (0.14, 0.58)) though no significant difference was noted for any FEV$_1$ values. Significant sex differences were also noted for FEV$_1$/FVC ratio among patients with CADs, with males showing values -0.07 (-0.12, -0.02) (p = 0.007) less than females' post- bronchodilator, with the pre- bronchodilator differences approaching significance as well (-0.05 (-0.1, 0), p = 0.057).

Further, a multivariate linear regression for differences in effect of CAD findings on PFTPs between sexes after adjustment for age, height, weight and smoking status using patients with no CAD radiology findings as reference was analysed (Table 5). Study patients displaying radiological evidence of CADs demonstrated significantly reduced absolute and percent predicted values for all PFTPs aside from BDR. Males showed a significantly greater effect of radiology abnormality with CADs on FEV$_1$ pre- (-0.53 (-0.74, -0.32) vs. -0.29 (-0.42, -0.16), p = 0.045) and post- (-0.51 (-0.72, -0.3) vs. -0.27 (-0.39, -0.14) p = 0.049) bronchodilator compared to females.

## Discussion

To the best of the authors' knowledge, this is the first study to examine sex differences in PFTPs among an Indigenous population, especially in the Australian Indigenous people from

**Table 3. Pulmonary function tests parameters according to presence of chronic airway disease radiological findings.**

| Radiology findings | | Without chronic airway disease | | | With chronic airway disease | | |
|---|---|---|---|---|---|---|---|
| | Sex | Female (n = 179) | Male (n = 111) | p-value | Female (n = 92) | Male (n = 103) | p-value |
| FVC | LLN (L) | 2.59 (2.71, 37.5) | 3.63 (3.83, 37.5) | <0.001* | 2.4 (2.58, 37.5) | 3.35 (3.55, 37.5) | <0.001* |
| | Pre (L) | 2.04 (2.23, 38.5) | 2.93 (3.24, 38.5) | <0.001* | 1.65 (1.88, 38.5) | 2.29 (2.58, 38.5) | <0.001* |
| | Pre (%) | 61.68 (66.42, 39.5) | 64.26 (69.76, 39.5) | 0.116 | 52.74 (59.41, 39.5) | 53.42 (59.61, 39.5) | 0.848 |
| | Post (L) | 2.1 (2.28, 40.5) | 2.98 (3.29, 40.5) | <0.001* | 1.75 (1.98, 40.5) | 2.42 (2.71, 40.5) | <0.001* |
| | Post (%) | 63.43 (67.95, 41.5) | 64.42 (70.48, 41.5) | 0.353 | 56.11 (62.52, 41.5) | 55.59 (62, 41.5) | 0.821 |
| | BDR (% change) | 2.05 (4.9, 42.5) | 0.49 (3.44, 42.5) | 0.166 | 3.93 (11.96, 42.5) | 4.78 (8.11, 42.5) | 0.477 |
| $FEV_1$ | LLN (L) | 2.03 (2.14, 43.5) | 2.82 (2.99, 43.5) | <0.001* | 1.87 (2.02, 43.5) | 2.56 (2.72, 43.5) | <0.001* |
| | Pre (L) | 1.56 (1.74, 44.5) | 2.2 (2.5, 44.5) | <0.001* | 1.08 (1.3, 44.5) | 1.37 (1.64, 44.5) | 0.001* |
| | Pre (%) | 57.93 (63.5, 45.5) | 60.36 (67.33, 45.5) | 0.168 | 43.58 (51.38, 45.5) | 40.69 (47.99, 45.5) | 0.244 |
| | Post (L) | 1.64 (1.81, 46.5) | 2.29 (2.59, 46.5) | <0.001* | 1.16 (1.39, 46.5) | 1.47 (1.75, 46.5) | <0.001* |
| | Post (%) | 60.87 (66.31, 47.5) | 62.68 (69.65, 47.5) | 0.250 | 46.72 (54.73, 47.5) | 43.85 (51.29, 47.5) | 0.253 |
| | BDR (% change) | 4.48 (7.63, 48.5) | 2.84 (6.42, 48.5) | 0.251 | 5.42 (11.25, 48.5) | 5.95 (10.23, 48.5) | 0.890 |
| $FEV_1$/FVC | LLN (absolute) | 0.69 (0.73, 49.5) | 0.69 (0.7, 49.5) | 0.070 | 0.69 (0.7, 49.5) | 0.67 (0.69, 49.5) | <0.001* |
| | Pre (absolute) | 0.75 (0.78, 50.5) | 0.73 (0.78, 50.5) | 0.453 | 0.63 (0.69, 50.5) | 0.57 (0.63, 50.5) | 0.004* |
| | Pre (%) | 91.55 (95.89, 51.5) | 92.14 (97.97, 51.5) | 0.463 | 78.97 (86.83, 51.5) | 73.08 (80.2, 51.5) | 0.020* |
| | Post (absolute) | 0.76 (0.79, 52.5) | 0.75 (0.79, 52.5) | 0.563 | 0.64 (0.7, 52.5) | 0.58 (0.64, 52.5) | 0.004* |
| | Post (%) | 93.7 (97.97, 53.5) | 94.56 (100.05, 53.5) | 0.403 | 79.62 (87.54, 53.5) | 74.23 (81.73, 53.5) | 0.043* |

Data were reported as mean (95% CI).

p-values derived from 2-tailed Students t-test.

*Denotes statistically significant association (p<0.05).

**Abbreviations**: **BDR**, Bronchodilator responsiveness; **$FEV_1$**, Forced expiratory volume in one second; **FVC**, Forced vital capacity; **LLN**, Lower limit of normal; **Post**: Post bronchodilator, **Pre**, Pre bronchodilator; **%**, percentage of predicted value.

the NT of Australia. Our study demonstrates that Indigenous Australian adult patients display significant sex differences in a number of PFTPs. Among patients with or without significant chest radiographic abnormality of CADs: *(i)* Females had reduced absolute values for FVC and $FEV_1$, but preserved percentage predicted values in comparison to their male counterparts; *(ii)* The $FEV_1$/FVC did not significantly differ between sex in either absolute or percent predicted values without CADs; *(iii)* Although, $FEV_1$/FVC values were lower for both sexes with underlying CAD in the multivariate analysis, $FEV_1$/FVC values was much more lower in males compared to females in the presence of underlying CADs; *(iv)* Males with radiological abnormality of CADs demonstrated significantly greater reductions in $FEV_1$ compared to females, and *(v)* No significant BDR was noted irrespective of whether the patient displayed chest radiological abnormalities or not.

Currently, normative reference equations for adult Indigenous people are lacking [24]. Hence, our study may be of significant value in characterising sex differences in PFTPs amongst the Indigenous population and in making ways in establishing normative reference values in the near future amongst this population. Moreover, two recently published studies comparing Global Lung function Initiative (GLI-2012) spirometric norms in adult Indigenous Australians demonstrated that FVC and $FEV_1$ do not correlate to any ethnic GLI groups [25] and no PFTPs fit the GLI-2012 regardless of which ethnic group was chosen, including "others/mixed" [19].

As observed in this study, sex differences in the PFTPs have been noted in other ethnic populations, overall displaying lower values among females in comparison to males [8–10]. Our

**Table 4. Multivariate linear regression for differences in pulmonary function test parameters (PFTPs) between sexes after adjustment for age, height, weight, and smoking status using female patient results as reference for CAD and no-CAD radiological findings.**

| Radiology findings | | Without CAD (n = 281) | | With CAD (n = 195) | |
|---|---|---|---|---|---|
| PFTPs | | β-coefficient | p-value | β-coefficient | p-value |
| FVC | LLN (L) | 0.49 (0.42, 0.57) | <0.001* | 0.47 (0.38, 0.56) | <0.001* |
| | Pre (L) | 0.41 (0.21, 0.62) | <0.001* | 0.33 (0.11, 0.56) | 0.004* |
| | Pre (%) | 0.57 (-4.71, 5.85) | 0.832 | -0.47 (-6.43, 5.5) | 0.878 |
| | Post (L) | 0.4 (0.2, 0.59) | <0.001* | 0.36 (0.14, 0.58) | 0.002* |
| | Post (%) | -0.41 (-5.71, 4.89) | 0.880 | -2.58 (-8.63, 3.46) | 0.400 |
| | BDR (% change) | -1.46 (-4.52, 1.6) | 0.348 | 0.83 (-4.6, 6.27) | 0.763 |
| $FEV_1$ | LLN (L) | 0.41 (0.33, 0.48) | <0.001* | 0.31 (0.25, 0.38) | <0.001* |
| | Pre (L) | 0.27 (0.07, 0.47) | 0.009* | 0.13 (-0.07, 0.34) | 0.193 |
| | Pre (%) | -0.06 (-6.45, 6.34) | 0.987 | -2.38 (-8.95, 4.19) | 0.476 |
| | Post (L) | 0.28 (0.08, 0.48) | 0.006* | 0.11 (-0.09, 0.32) | 0.278 |
| | Post (%) | -0.34 (-6.63, 5.95) | 0.916 | -3 (-9.72, 3.72) | 0.380 |
| | BDR (% change) | -0.58 (-4.06, 2.9) | 0.743 | -2 (-6.72, 2.75) | 0.408 |
| $FEV_1/FVC$ | LLN (absolute) | -0.02 (-0.06, 0.01) | 0.174 | -0.02 (-0.02, -0.02) | <0.001* |
| | Pre (absolute) | -0.02 (-0.06, 0.02) | 0.306 | -0.05 (-0.1, 0) | 0.057 |
| | Pre (%) | 1.12 (-4, 6.23) | 0.668 | -4.06 (-10.46, 2.33) | 0.212 |
| | Post (absolute) | -0.01 (-0.05, 0.03) | 0.588 | -0.07 (-0.12, -0.02) | 0.007* |
| | Post (%) | 2.21 (-2.77, 7.19) | 0.383 | -5.65 (-12, 0.71) | 0.081 |

Data were reported as mean (95% CI). p-values derived from factorial effect of sex in the multivariate regression.

*Denotes statistically significant association (p<0.05).

**Abbreviations**: **BDR**, Bronchodilator responsiveness; **CAD**: Chronic airway disease; **$FEV_1$**, Forced expiratory volume in one second; **FVC**, Forced vital capacity; **LLN**, Lower limit of normal; **Post**: Post bronchodilator, **Pre**, Pre bronchodilator; **%**, percentage of predicted value.

study confirms for the first time that this is indeed true for Australian Indigenous females to demonstrate lower PFTPs compared to their male counterparts. Typically, it is observed that males have larger values for FVC and $FEV_1$, with no significant difference in the $FEV_1/FVC$ ratio or tend to have reduced $FEV_1/FVC$ ratio compared to females. This is likely related to "dysanapsis", the differential growth between airway and lung size [2–4]. Due to these underlying sex differences, it is plausible that the effects of disease on PFTPs will also differ. These differences may also be related to other internal or external factors as previously documented in the published literature [2–10, 26–29]. However, it was beyond the scope of this current study to explore other potential factors contributing to sex differences in PFTPs in our study patients.

We observed a greater reduction in percentage predicted values among males compared to females in the presence of CADs; FVC 10 vs. 6%, $FEV_1$ 15 vs. 11%, $FEV_1/FVC$ 13 vs. 10%. Although these differences did not reach statistical significance in the current study, it is plausible they may be clinically relevant. A previous study in the same setting and population also demonstrated significant sex differences in the clinical manifestations of patients with obstructive sleep apnoea [30]. Further longitudinal studies are warranted to document the long-term pattern of PFTPs in this Indigenous population, as previous reports indicate sex may have an impact on decline in PFTPs, with females demonstrating earlier decline in comparison to their male counterparts [31].

Spirometry is a simple and useful tool that can be utilised in the accurate diagnosis and management of respiratory disorders in day-to-day clinical practice even at the primary health

**Table 5. Multivariate linear regression for differences in effect of CAD radiology findings on pulmonary function test parameters (PFTPs) between sexes after adjustment for age, height, weight, and smoking status using patients with no CAD radiology findings as reference.**

| | Sex | Females (n = 265) | | Males (n = 211) | | Difference |
|---|---|---|---|---|---|---|
| **PFTPs** | | β-coefficient | ^p-value | β-coefficient | ^p-value | #p-value |
| **FVC** | Pre (L) | -0.2 (-0.34, -0.06) | 0.005* | -0.4 (-0.61, -0.18) | <0.001* | 0.111 |
| | Pre (%) | -6.06 (-10.3, -1.81) | 0.005* | -9.56 (-14.39, -4.72) | <0.001* | 0.263 |
| | Post (L) | -0.15 (-0.28, -0.02) | 0.021* | -0.3 (-0.52, -0.08) | 0.007 | 0.228 |
| | Post (%) | -4.67 (-8.71, -0.62) | 0.024* | -7.35 (-12.53, -2.18) | 0.006* | 0.408 |
| | BDR (% change) | 3.29 (-0.39, 6.97) | 0.079 | 4.4 (1.85, 6.95) | 0.001* | 0.590 |
| **FEV$_1$** | Pre (L) | -0.29 (-0.42, -0.16) | <0.001* | -0.53 (-0.74, -0.32) | <0.001* | 0.045* |
| | Pre (%) | -10.85 (-15.81, -5.89) | <0.001* | -15.31 (-21.12, -9.5) | <0.001* | 0.241 |
| | Post (L) | -0.27 (-0.39, -0.14) | <0.001* | -0.51 (-0.72, -0.3) | <0.001* | 0.049* |
| | Post (%) | -10.15 (-15.07, -5.23) | <0.001* | -14.27 (-20.11, -8.43) | <0.001* | 0.284 |
| | BDR (% change) | 2.57 (-0.58, 5.73) | 0.110 | 2.82 (-0.45, 6.09) | 0.091 | 0.918 |
| **FEV$_1$/FVC** | Pre (absolute) | -0.08 (-0.12, -0.05) | <0.001* | -0.1 (-0.14, -0.06) | <0.001* | 0.438 |
| | Pre (%) | -10.09 (-14.43, -5.75) | <0.001* | -12.77 (-17.9, -7.64) | <0.001* | 0.426 |
| | Post (absolute) | -0.08 (-0.11, -0.05) | <0.001* | -0.12 (-0.16, -0.08) | <0.001* | 0.187 |
| | Post (%) | -10.51 (-14.81, -6.21) | <0.001* | -14.06 (-19.26, -8.87) | <0.001* | 0.310 |

Continuous data were reported as mean (95% CI).

^p-values derived from factorial effect of radiology abnormality in the multivariate regression.

#p-value derived from post- estimation test of differences in β-coefficients between sexes.

*Denotes statistically significant association (p<0.05).

**Abbreviations**: **BDR**, Bronchodilator responsiveness; **CAD**, Chronic airway disease; **FEV$_1$**, Forced expiratory volume in one second; **FVC**, Forced vital capacity; **LLN**, Lower limit of normal; **Post**: Post bronchodilator, **Pre**, Pre bronchodilator; **%**: Percentage of predicted value.

care level. However, lack of awareness of sex differences in the PFTPs may lead to inaccurate diagnoses [5–7]. In this study, we observed females displayed significantly reduced values for FVC and FEV$_1$ although the percent predicted values did not show any statistically significant difference between the sexes either with or without radiological evidence of CADs. While FEV$_1$/FVC did not demonstrate significant differences between sexes for study patients without CADs, it did so for patients with radiological evidence of CADs. This appears largely driven by the observed larger reduction in FEV$_1$ in the presence of CADs for males compared to females (-0.53 vs. -0.29 L, respectively). This stronger effect of disease on FEV$_1$ in males may be a result of the relatively reduced growth in airways compared to lung parenchyma/alveoli or alternatively due to higher smoking rates or age-related decline in FEV$_1$ or increasing severity of the underlying disease [2–10, 26, 31, 32].

Indigenous people have a higher burden of chronic health conditions, including cardio-respiratory disorders, giving rise to higher morbidity and mortality [33–37]. Understanding the different clinical manifestations [38–40] and appropriate interventions [41–50] will help in early diagnosis and management of chronic health conditions in the Indigenous population, for better health related outcomes. Varying manifestations of sex differences in PFTP's have been demonstrated in this study, both with and without underlying radiological evidence of CADs among an Indigenous Australian cohort. This could also have implications in classifying severity of CADs [51] and in the clinical decision making while considering airway directed inhaled pharmacotherapy [52]. We believe the results represented in this study may be an avenue or encourage other researchers in characterising sex differences in other Indigenous populations, with a view to establishing normative reference lung function values for adult Indigenous population.

## Limitations

The results of this study are restricted to the Indigenous study patients from the TEHS region of the NT of Australia. The results and outcomes may not be applicable to other Indigenous populations. As detailed in our previous report [19], we acknowledge the fact that in this study, among patients with no radiological evidence of CADs, one may not be able to exclude presence of disease, especially with a smoking history. The correlation of respiratory symptoms to chest radiology could be variable, especially with CADs (symptomatic with normal radiology/asymptomatic with abnormal radiology). While normal chest radiology may exclude significant parenchymal abnormalities, it may not entirely exclude chronic airway pattern. Moreover, we did not assess the severity of CADs according to radiology that may have led to some bias in the outcomes represented in this study. Nevertheless, this is the first study to document sex differences in the PFTPs amongst an Indigenous cohort and there is scope for further research.

## Conclusion

Currently there are no PFT norms or sex differences available specific to the healthy adult Indigenous Australian population. In this study, we observed, in comparison to males, that adult Indigenous Australian females have a tendency towards lower PFTPs values for absolute FVC and $FEV_1$, but no significant difference for percentage predicted values irrespective of the presence or absence of radiological evidence of CADs. The $FEV_1/FVC$ predicted values are also likely to demonstrate no significant sex differences with no significant underlying CADs; however may display reduced values in both sexes with underlying CADs. Moreover, males with radiological abnormality tend to demonstrate greater reductions in $FEV_1$ compared to females.

## Acknowledgments

We sincerely thank all the respiratory technologists and Respiratory Clinical Nurse Consultants from Darwin Respiratory and Sleep health and Royal Darwin Hospital, Darwin Private Hospital, Darwin, Australia, for their invaluable contribution towards this study. We thank Mr. Xinlin Jing, Health Information Services, Royal Darwin Hospital, Darwin, Northern Territory, Australia for helping with data collection for this study. We also extend our sincere gratitude to our research assistant, Mrs. Joy J Mingi, Department of Public Health, Charles Darwin University, Darwin, Northern Territory, Australia and special thanks to Ms. Ara Joy Perez from Darwin Respiratory and Sleep health, Darwin Private Hospital, Darwin, Australia for her invaluable contribution towards this study. We also extend our sincere appreciation to our Indigenous health workers, especially to Mr Izaak Thomas (Australian Indigenous Luritja descendent) from the chronic respiratory disease co-ordination division in reviewing and approving this manuscript for the appropriateness of the representation and respect to the Indigenous context represented in this study. Finally we thank Ms Elisha White, senior respiratory scientist for her contribution towards this study.

## Author Contributions

**Conceptualization:** Subash S. Heraganahally, Timothy Howarth, Lisa Sorger, Helmi Ben Saad.

**Data curation:** Subash S. Heraganahally, Timothy Howarth.

**Formal analysis:** Subash S. Heraganahally, Timothy Howarth.

**Investigation:** Subash S. Heraganahally, Timothy Howarth, Helmi Ben Saad.

**Methodology:** Subash S. Heraganahally, Timothy Howarth, Helmi Ben Saad.

**Project administration:** Subash S. Heraganahally.

**Resources:** Subash S. Heraganahally.

**Software:** Timothy Howarth.

**Supervision:** Subash S. Heraganahally, Helmi Ben Saad.

**Validation:** Subash S. Heraganahally, Timothy Howarth, Lisa Sorger, Helmi Ben Saad.

**Visualization:** Subash S. Heraganahally, Timothy Howarth, Lisa Sorger, Helmi Ben Saad.

**Writing – original draft:** Subash S. Heraganahally, Timothy Howarth, Helmi Ben Saad.

**Writing – review & editing:** Subash S. Heraganahally, Timothy Howarth, Lisa Sorger, Helmi Ben Saad.

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
