## [Decision Letter · Decision Letter 0]

27 Dec 2021

PONE-D-21-26363Sex differences in pulmonary function parameters among Indigenous Australians with and without chronic airway diseasePLOS ONE

Dear Dr. Heraganahally,

Thank you for submitting your manuscript to PLOS ONE. After careful consideration, we feel that it has merit but does not fully meet PLOS ONE’s publication criteria as it currently stands. Therefore, we invite you to submit a revised version of the manuscript that addresses the points raised during the review process.

In the interests of time, I've reviewed the manuscript myself (had a problem with a late reviewer), and I agree with the first reviewer. However, these are relatively major changes - the paper needs a power analysis, inclusion criteria, and a pass through for readability.

We look forward to receiving your revised manuscript.

Kind regards,

James West, PhD

Academic Editor

PLOS ONE

Journal Requirements:

Additional Editor Comments (if provided):

In the interests of time, I've reviewed the manuscript myself (had a problem with a late reviewer), and I agree with the first reviewer. However, these are relatively major changes - the paper needs a power analysis, inclusion criteria, and a pass through for readability.

Reviewers' comments:

Reviewer's Responses to Questions

**Comments to the Author**

1. Is the manuscript technically sound, and do the data support the conclusions?

Reviewer #1: Yes

2. Has the statistical analysis been performed appropriately and rigorously? 

Reviewer #1: Yes

3. Have the authors made all data underlying the findings in their manuscript fully available?

Reviewer #1: Yes

4. Is the manuscript presented in an intelligible fashion and written in standard English?

Reviewer #1: Yes

5. Review Comments to the Author

Reviewer #1: the author performed a retrospective study on 485 adults to assess Sex differences in pulmonary function parameters among Indigenous Australians with and without chronic airway disease and concluded that There are significant sex differences in the PFTPs in Indigenous Australians. Recognizing these differences may be of value in the accurate diagnosis, management, monitoring and prognostication of CADs in this population. the topic is important and the manuscript is well designed and written but i have few comments:

1- the manuscript needs good English editing

2- what were the inclusion and the exclusion criteria?

3- how did you estimate the sample size for this study? what was the power of the study?

6. PLOS authors have the option to publish the peer review history of their article (what does this mean?). If published, this will include your full peer review and any attached files.

Reviewer #1: **Yes: **Doaa El Amrousy

---

## [Author Response · Author response to Decision Letter 0]

8 Jan 2022

We sincerely appreciate the reviewers’ time, comments and suggestions regarding our study. We have made changes to the revised manuscript taking into consideration of the reviewers’ suggestions and comments, which are detailed below.

Response to reviewers comments.

Reviewer: 1

Comments to the Author

Reviewer #1: the author performed a retrospective study on 485 adults to assess Sex differences in pulmonary function parameters among Indigenous Australians with and without chronic airway disease and concluded that There are significant sex differences in the PFTPs in Indigenous Australians. Recognizing these differences may be of value in the accurate diagnosis, management, monitoring and prognostication of CADs in this population. the topic is important and the manuscript is well designed and written but i have few comments:

Response: We sincerely thank and very pleased for the reviewer above opinion and comments. Especially for the comment that “the topic is important and the manuscript is well designed and written”

1- the manuscript needs good English editing

Response: We thank the reviewer for bring this to our attention. We have addressed this in the revised manuscript were required and appropriate. 

2- what were the inclusion and the exclusion criteria?

Response: We thank the reviewer for asking us to clarify regarding this. We have addressed this in the revised manuscript and reads as in the revised manuscript under the method section

Study participants – Inclusion and exclusion criteria

Patients who underwent PFTs and had a chest radiography available (Chest X-Ray and computed tomography (CT)) to assess the presence/absence of any radiological pulmonary abnormalities (specifically demonstrating radiographic evidence of COPD/emphysema, bronchiectasis or a combination of both COPD & bronchiectasis) were included in the analysis. Patients PFTs that were assessed not acceptable for session quality were excluded.

3- how did you estimate the sample size for this study? what was the power of the study?

Response: We thank the reviewer for asking us to clarify regarding this. We have added a subsection entitled “sample size” . 

As it was a retrospective study, we did not calculate sample size. 

In the revised manuscript under the subheading - Statistical analysis it reads as 

“As the study was a retrospective study no prior sample size estimation was conducted, however, post-hoc power analysis was conducted for testing of mean differences between females and males for pre- bronchodilator values of FVC, FEV1 and FEV1/FVC (Table1).”

However, we did calculate a sample size for this study during our revision as per the reviewer suggestion as detailed below, but did not included in the revised manuscript. The authors are in the opinion that the above statement may be appropriate for this study. If the reviewer like us to include the below statement, we are happy to consider. 

“Given the pioneer character of our study, these two values were obtained from a comparative study of PFTPs in healthy male (n=75) and female (n=75) subjects (age range: 20-75 years) of western Rajasthan in India 9, where FEV1/FVC means±SD of females and males were, respectively, 0.8780±0.0818 and 0.8516±0.0813 (ie; a common standard-deviation equal to 0.0815). 

The sample size for the study was 407 (230 females and 177 males). The assumption of a percentage of 70% for exclusion criteria gave a revised sample of 1357 patients [= 407/(1-0.70)].”

We hope that we have addressed the reviewers comments and suggestion satisfactorily and are acceptable to the reviewer and the editor.

---

## [Decision Letter · Decision Letter 1]

26 Jan 2022

Sex differences in pulmonary function parameters among Indigenous Australians with and without chronic airway disease

PONE-D-21-26363R1

Dear Dr. Heraganahally,

We’re pleased to inform you that your manuscript has been judged scientifically suitable for publication and will be formally accepted for publication once it meets all outstanding technical requirements.

Kind regards,

James West, PhD

Academic Editor

PLOS ONE

Additional Editor Comments (optional):

Reviewers' comments:

Reviewer's Responses to Questions

**Comments to the Author**

1. If the authors have adequately addressed your comments raised in a previous round of review and you feel that this manuscript is now acceptable for publication, you may indicate that here to bypass the “Comments to the Author” section, enter your conflict of interest statement in the “Confidential to Editor” section, and submit your "Accept" recommendation.

Reviewer #1: All comments have been addressed

2. Is the manuscript technically sound, and do the data support the conclusions?

Reviewer #1: Yes

3. Has the statistical analysis been performed appropriately and rigorously? 

Reviewer #1: Yes

4. Have the authors made all data underlying the findings in their manuscript fully available?

Reviewer #1: Yes

5. Is the manuscript presented in an intelligible fashion and written in standard English?

Reviewer #1: Yes

6. Review Comments to the Author

Reviewer #1: The authors performed the required changes. The manuscript improved and The manuscript now is ready for publication.

7. PLOS authors have the option to publish the peer review history of their article (what does this mean?). If published, this will include your full peer review and any attached files.

Reviewer #1: **Yes: **Doaa El Amrousy

---

## [Editor Report · Acceptance letter]

31 Jan 2022

PONE-D-21-26363R1 

Sex differences in pulmonary function parameters among Indigenous Australians with and without chronic airway disease 

Dear Dr. Heraganahally:

I'm pleased to inform you that your manuscript has been deemed suitable for publication in PLOS ONE. Congratulations! Your manuscript is now with our production department. 

Kind regards, 

on behalf of

Dr. James West 

Academic Editor

PLOS ONE